# Laser-Based People Detection and Obstacle Avoidance for a Hospital Transport Robot

**DOI:** 10.3390/s21030961

**Published:** 2021-02-01

**Authors:** Kuisong Zheng, Feng Wu, Xiaoping Chen

**Affiliations:** School of Computer Science and Technology, University of Science and Technology of China, Hefei 230026, China; kszheng@mail.ustc.edu.cn (K.Z.); xpchen@ustc.edu.cn (X.C.)

**Keywords:** service robot, navigation, people detection, obstacle avoidance

## Abstract

This paper describes the development of a laser-based people detection and obstacle avoidance algorithm for a differential-drive robot, which is used for transporting materials along a reference path in hospital domains. Detecting humans from laser data is an important functionality for the safety of navigation in the shared workspace with people. Nevertheless, traditional methods normally utilize machine learning techniques on hand-crafted geometrical features extracted from individual clusters. Moreover, the datasets used to train the models are usually small and need to manually label every laser scan, increasing the difficulty and cost of deploying people detection algorithms in new environments. To tackle these problems, (1) we propose a novel deep learning-based method, which uses the deep neural network in a sliding window fashion to effectively classify every single point of a laser scan. (2) To increase the speed of inference without losing performance, we use a jump distance clustering method to decrease the number of points needed to be evaluated. (3) To reduce the workload of labeling data, we also propose an approach to automatically annotate datasets collected in real scenarios. In general, the proposed approach runs in real-time and performs much better than traditional methods. Secondly, conventional pure reactive obstacle avoidance algorithms can produce inefficient and oscillatory behaviors in dynamic environments, making pedestrians confused and possibly leading to dangerous reactions. To improve the legibility and naturalness of obstacle avoidance in human crowded environments, we introduce a sampling-based local path planner, similar to the method used in autonomous driving cars. The key idea is to avoid obstacles by switching lanes. We also adopt a simple rule to decrease the number of unnecessary deviations from the reference path. Experiments carried out in real-world environments confirmed the effectiveness of the proposed algorithms.

## 1. Introduction

Due to the large volume of goods transported every day, modern hospitals have a great demand for logistics automation systems, to reduce the cost associated with manual deliveries, increase efficiency in material transport and improve service quality [1,2]. Nowadays, mobile robots have been considered to be one of the most feasible solutions for automation of transportation in hospitals, due to their flexibility and the need for minor modification in infrastructures compared to other systems, such as pneumatic tube systems and electric track vehicle systems [1,3]. Therefore, many research efforts have been devoted to designing a robot-based logistic system, and several delivery robots have been developed and tested in realistic scenarios over the past thirty years [4,5,6,7,8]. However, there are still various challenges that need to be considered when deploying autonomous robots in human-populated environments such as hospitals. In this work, we focus mainly on people detection with laser scanners and obstacle avoidance in crowds of people because they are keys to the success of our application.

People detection is an important functionality for robots that share workspaces with people as in a hospital environment. Laser range finders commonly mounted on mobile robots are an appropriate sensor for this task since they are less susceptible to ambient light and provide rapid and precise distance measurements over a wide field of view. However, the information contained in a single 2D range scan is considered not sufficient to reliably distinguish humans from other objects in cluttered environments. Thus, several approaches [9,10,11] adopt multi-layer or 3D laser scanners to improve the robustness of classification.

Traditional approaches [12,13,14,15,16] for people detection with a single-layer laser scanner roughly consists of three steps: segmentation, feature extraction, and classification. Detection with multi-layer laser scanners are similar to the single-layer cases, except an additional fusion step which combines the hypothesis of different layers into the final detection [9,10,17,18]. Traditional approaches come with several inherent drawbacks: (1) the jump distance threshold can lead to oversegmentation; (2) the hand-crafted features may be suboptimal; (3) the geometrical features are extracted only from individual segments; (4) body parts found in each layer are combined using a heuristic person shape model.

Recently, Beyer et al. [19,20] first applied convolutional neural networks (CNNs) for object detection in 2D laser data, eliminating the need for feature engineering and enabling drastic improvements. To address the spatial density problem of laser data, they proposed a preprocessing stage, which cuts out and normalizes a fixed real-world extent window around each laser point by making use of the spatial information that a laser sensor provides.

In this paper, we present a deep learning-based detector for detecting humans using laser range finders mounted on a mobile robot. This is done by assigning a label (person or nonperson) to each laser point in a laser scan (as shown in the video). The key idea is to evaluate PointNet [21] in a sliding window fashion. To improve speed during inference, the laser scan is split into segments via jump distance clustering algorithm [13]. Then the circle (cylindrical) neighborhoods of centroids of each segment are inputted to PointNet for classification. Points from the same segments are considered belonging to the same category. By doing so, our detector runs in real-time and performs well in populated environments. Furthermore, we introduce an approach to automatically annotate datasets collected in real scenarios. As the input of the network is a point-cloud, the proposed method can also be adapted for detection from data produced by 3D lasers or multi-layer laser scanners.

The points belonging to humans are published to the proxemic layer, which alters the local cost map [22] with a Gaussian distribution around each of these points to increase the cost near people, leading to more comfortable and safe navigation by encouraging the robot to keep a *social distance* to surrounding humans [23]. The detection results can also be used to adapt the robot’s velocity according to its distance to people.

The basic task of our transport robot is to deliver materials from the start position to the goal position along the global path generated by the global planner using an a priori roadmap. To accomplish this task, the robot requires to be able to follow the path while simultaneously avoid obstacles. However, it is still a very challenging problem to safely avoid obstacles in dynamic environments populated with people due to the unpredictable motion of humans. Many factors have an impact on the performance, such as human comfort, naturalness as well as social rules [23]. To ensure that the robot does not make humans feel uncomfortable or threatened, the avoidance algorithm must satisfy the following requirements of our application:The behavior produced should be legible and predictable by humans, to improve navigation efficiency and avoid dangerous people reactions [24];The maximum deviation from the reference path should be limited within some upper bound [25,26];The trajectory generated by the local planner should lead to the target point.

Several studies have studied this task and proposed different algorithms, which are implemented in real robots [6,8,26]. Evans et al. [6] propose the BUG2 method which detects the boundary of obstacles and guide the robot around obstacles by following their boundary. Sagobissa and Zaccaria [26] introduce Roaming Trails to allow the robot to use Artificial Potential Fields (APFs) to avoid obstacles in a constrained (diamond-shaped) area around the global path, which reduce undesired behavior in crowded environments and guarantees the robot can never be trapped in deadlocks. However, the robot’s movement can be confused in dynamic environments and they do not explicitly take account of human comfort. Takahashi et al. [8] propose the repulsive potential function of APF with a time scale as a design parameter. However, it possibly gets in the trap of local minima.

We adopt a state-space sampling-based path planner, which has been successfully applied in autonomous driving vehicles in road driving scenario [27,28,29,30,31]. The core idea is to laterally sample multiple path candidates around the reference path, andthen select the best one by maximizing an objective function, which includes a measure of the proximity to the nearby obstacles, social norms (passing on the right side) and the deviation from the reference path. To ensure the robot to reach the target, all sampled paths are converged to the same endpoint as the original path. Unlike many reactive methods [32,33,34,35], which generate wavy trajectories, our method produces straight paths and the robot’s heading aligns with the reference path most of the time. Thus the robot’s behavior is easy to predict and its intention can be intuitively understood by nearby people. Besides, our method is easy to implement and takes little computation resources.

## 2. Related Work

A variety of research works have showed that service robots can perform navigation-related tasks in populated environments, such as guiding tourists in museums [36], leadingthe way for customers in shopping malls [37] andconveying various items in hospitals [4,38]. Apart from research robots, several commercial robots are developed and deployed in a few hospitals. Related works about each specific aspect of our system are described in the following subsections.

### 2.1. People Detection in Laser Data

Several previous works focus on people/leg detection from 2D laser data. The most primitive approaches detect human legs based on the size of clusters [39,40,41]. Mendes et al. [16] join small segments separated by a distance less than 50 cm into one segment (legs) and introduce a voting scheme, which considers hypotheses over time to classify an object with high confidence. Fod et al. [42] filter the range measurements to remove background and group adjacent foreground reading as blobs which are tracked via the Kalman Filter, the moving blobs are assumed as people. Zhao and Shibasaki [43] use a similar method for detecting people with multiple single-row laser range scanners. Schulz et al. [44,45] detect objects as local minima in the distance profile of a range scan, the changes in consecutive scans are also considered to distinguish between static and moving objects. Cui et al. [46] identify legs as the local maxima in the accumulated distribution of successive laser frames which are subtracted by the background image.

The performance of recognition can be improved by replacing the hand-tuned thresholds with machine learning techniques. Arras et al. [12] first utilize the AdaBoost algorithm to train a strong classifier from 14 real value features extracted from groups of neighboring beams. Spinello and Siegwart [15] propose a graph cutting method to alleviate the oversegmentation problem caused by the classic jump distance method. Weinrich et al. [13] develop a detector based on generic distance-invariant feature for people detection and the distinction of their walking aids. Themethods mentioned above apply the AdaBoost classifier on hand-crafted features. Chung et al. [47] inductively derive the common attributes of legs from a large number of sample data and train support vector domain description (SVDD) [48] on 3 simple attributes to detect legs. Recently Beyer et al. [19,20] show how to apply CNNs on 2D range data to efficiently and effectively detect wheelchairs/walkers.

Several works take advantage of multi-layer laser sensors to improve classification performance and reduce false detections. Mozos et al. [9] introduce a method of combining the hypotheses in different layers. Gidel et al. [17] use Parzen methods to detect pedestrian positions of each laser plane, which are then sent to a decentralized fusion according to the 4 planes. Kim et al. [10] propose the adaptive breakpoint detector (ABD) segmentation method and 15 new features for 2.5D laser data and employ the radial basis function additive kernel support vector machine technique for classification to reduce the computation time while maintaining the performance.

Recently, several works concentrate on deep learning methods for point cloud understanding tasks. Qi et al. introduce an efficient type of neural network named PointNet [21], which directly consumes unordered point sets and obtains state-of-the-art performance. To achieve permutation invariance, PointNet uses a symmetric aggregate function on the pointwise features generated by a shared Multi-Layer Perceptron (MLP). PointNet has strong representation ability but does not capture local structures. To recognize fine-grained patterns, PointNet++ [49] adopts a hierarchical architecture that applies PointNet recursively on a nested partitioning of the input point set. Several studies employ PointNet to encode points in grid cells and integrate 2D or 3D CNNs for object detection [50,51]. However, these methods are too intricate and not effective for our tasks.

### 2.2. Local Obstacle Avoidance

Obstacle avoidance or local path planning has been studied in a large number of works. The most common approaches refer to pure reactive ones which calculate the heading direction or velocity command based only on local sensory information [13,32,52,53]. For example, the Vector Field Histogram (VFH) [52] method models the local world with a two-dimensional Cartesian histogram grid, which is reduced to a one-dimension polar histogram containing the polar obstacle density in that direction, then selects the histogram sectors with a low polar obstacle density as the steering direction. TheCurvature-Velocity Method (CVM) method [53] takes account of robot limitations and environment constraints, it formulates the problem as one of constrained optimization in the velocity space. However, pure reactive methods mentioned above have several drawbacks in dynamic environments: (1) they are local methods and potentially lead the robot to bad situations; (2) they can produce undesired behavior in dynamic environments; (3) the trajectories generated in complex situations are hard to predict and may conflict with the social rules followed by humans.

Several extensions have been developed to address the situations that are problematic for pure reactive methods [33,34,37,54]. VFH* [33] combines VFH with A* search algorithm for look-ahead verification. Ratering and Gini [54] propose a hybrid artificial potential field combining a global discontinuous potential field and local continuous potential field to alleviate the local minima problem. The Forbidden Velocity [34] generalizes the Dynamic Window Approach (DWA) to consider moving obstacles.

Human comfort is considered in a lot of literature [23,55,56,57,58]. Hall [59] first introduced the idea of the personal space called proxemics (the invisible bubble of space around people) that people like to keep between themselves. The personal space is modeled as a cost function which is used for human-aware navigation [55,56,57]. Shi et al. [58] develop human-aware velocity constraints as a function of the distance of the robot from a human. Kruse et al. [23] experimentally examine how humans deal with pass crossing and provide a context-dependent social cost for legible robot navigation.

State-space sampling-based methods are often used in highly constrained environments, compared to control space sampling-based method, they explicitly take both kinodynamic and environmental constraints into account, leading to an efficient sampling scheme. Thrun et al. [27] draw candidate path from a 2D space of maneuvers, i.e., lateral offset and changing rate. Howard and Ferguson [28] present a model-based trajectory generation approach for state-space sampling. Werling et al. [13] propose a method to sampling trajectories utilizing optimal-control strategies within the Frenet-Frame.

## 3. System Overview

### 3.1. Robot Design

Figure 1 shows the prototype differential drive robot with a width of 50 cm, a length of 70 cm and a height of 120 cm. The robot weighs about 50 kg and has a maximum load of about 100 kg. When the robot is fully charged, it can run continuously for 5 h at 20% of the maximum load. The maximum slope angle that the robot can climb is about $10^{\circ }$ when fully loaded. To improve the safety of navigation and the quality of localization, the robot is equipped with two Hokuyo UTM-30LX laser range scanners, each laser scanner operates at a frequency of 40 Hz, has a field of view $270^{\circ }$ and a maximum detection range of 30 m. The data of the two laser range scanners are merged to cover a field of view $360^{\circ }$ with the ira_laser_tools package. The robot is equipped with a touch screen to allow an operator to interact with it. The robot is also equipped with cameras and a stereo camera, they can be used to detect useful information, such as types of obstacle ahead, the orientation of people, but these functions will be done in future work.

### 3.2. System Architecture

The software architecture is shown in Figure 2. All modules are implemented on the Robot Operating System (ROS) [60], which provides a set of tools that help users build robot applications, such as the Adaptive Monte-Carlo Localization (AMCL) [61] and the layered cost map [22]. The basic task of the robot is to transport materials from the start location to the goal location. More specifically, when an operator enters the goal position, the global planner is called to search a global path from the start position to the destination with the topological graph. The n the robot navigates to the goal location along the global path while avoiding the static or dynamic obstacles using the local path planner. To improve the safety of navigation, the robot continuously detects people from laser data with the people detection module. The points detected as belonging to people are integrated into the local cost map for use by the local path planner. In this paper, we mainly concentrate on two components: the people detector and the local path planner. The purpose of this paper is to improve the safety and legibility of navigation in human-populated environment.

### 3.3. Maps

Both grid-based and topological maps are utilized to model the multi-floor indoor environment. Grid maps are built via a simultaneous localization and mapping algorithm (SLAM) [62] using the data collected from each floor of a building. Grid maps are used for localization since they provide accurate information about environments. Topological maps are used for path planning since they provide natural interfaces for human instructions and permit fast planning in large-scale environments. Topological maps are created manually based on the grid maps using a graphical tool. Nodes of graphs represent distinct places, such as doors, intersections or elevators. Topological graphs of multiple floors are connected together according to elevator names to generate a complete topological graph for use by the global path planner.

## 4. People Detection

The observation of a laser scanner consists of a sequence of beams $O=\{b_{1},\ldots ,b_{N}\}$. Each beam $b_{i}$ corresponds to a tuple $\left(\rho_{i},\phi_{i}\right)$, where $\rho_{i}$ is the length of the beam, $\phi_{i}$ is the angle of the beam. The polar coordinates are converted to Cartesian coordinates $S=\{x_{1},\ldots ,x_{N}\}$, $x_{i}=\left(x_{i},y_{i}\right)$, where $x_{i}=\rho_{i}\cos\left(\phi_{i}\right)$ and $y_{i}=\rho_{i}sin\left(\phi_{i}\right)$. The goal of our approach is to calculate the labels of all points of a laser scan.

### 4.1. Preprocessing

Since laser data have strong variations in point density, which naturally decrease quadratically with distance, the voxel-grid filter is employed to down-sample the points in high-density regions. The filter divides the 2D space into a regular grid of cells of a given resolution and projects laser points onto the grid, then it randomly selects only one point from each nonempty grid cell. Furthermore, the laser points which are 10 m away from the robot’s center are filtered out, since these points are too sparse to be used to reliably detect people.

### 4.2. Inputs

The input of the network is the resampled neighborhood of a proposal point, as shown in Figure 2. More specifically, for each proposal position $p_{i}$, we use a *k*D-tree to find all its neighbor points $N_{i}$ within the radius *r*. which are then centered around the current query point: $p_{i}^{\left(j\right)}-p_{i}$ for $p_{i}^{\left(j\right)}\in N_{i}$. To take advantage of the parallel computing of a GPU, all neighborhoods are uniformly resampled to a predefined size *K*. In other words, if a neighborhood contains too many points, it is uniformly resampled to the size *K*. Conversely, if a sample contains too few points, zero paddings are applied. Note that the inputs are inherently translation-invariant due to the use of relative coordinates. Additionally, to increase the robustness against rotations, we augment the dataset with random rotation within [0,2π) during training.

### 4.3. Network

The network architecture is illustrated in Figure 3. To be robust against rotation transformations, a mini-PointNet (T-Net) takes the input and regresses to a 2×2 matrix, which is multiplied by the input for 2D transformation. Unlike the original network, the input is only transformed in the X-Y plane. As hared MLP(64,128,1024) is utilized to project the transformed points into a high dimensional feature space. After that, a max-pooling operation is applied to aggregate the individual point features into the global feature, which are further processed by a fully connected neural network with the size 512,256,1. A dropout layer with a keeping ratio 0.7 is applied on the last fully connected layer. We use the ReLU activation function followed by batch normalization in each fully connected layer except the last one. The output layer uses the sigmoid function to produce the probability of the class that the input belongs to. The loss function is the binary cross-entropy loss.

### 4.4. Inference

To decrease the number of points needed to be classified during inference time, the jump distance clustering method is applied to split a laser scan into groups of beams. Only the point nearest to the centroid of a group is chosen to be evaluated, other points in the same cluster are assumed to belong to the same object. After clustering, the number of points needed to be classified is decreased and the inference speed can be accelerated to about 5 × faster.

The jump distance algorithm iterates over the range scan. If the difference of measurements of two adjacent beams $|\rho_{i}-\rho_{i-1}|$ is over a certain threshold δ, anew group is initialized there. The output of this partitioning procedure is an angular ordered sequence of segments, $S=\{S_{1},S_{2},\ldots ,S_{s}\}$. Decreasing the threshold δ will increase the number of the centers of clusters needed to be evaluated. The distance threshold is set to 0.1 m here, which is good enough to split human legs apart from the background. An example grouping results of a laser scan is illustrated in Figure 4.

### 4.5. Automated Dataset Annotation

It is a challenging and time-consuming task to manually annotate all laser scans in a large dataset. In this section, we describe the methods for dataset collection and automatic labeling.

There are two types of datasets needed to be collected: the dynamic datasets obtained by placing the robot in a site with significant pedestrian traffic (e.g., hallway), and the static datasets collected by joy-sticking the robot in an environment devoid of people. All laser points in a static dataset collected from a moving robot are simply treated as static obstacles.

To calculate the labels of the laser points in a dynamic dataset recorded from a stationary robot, we calculate the background of the dataset and subtract it to get the points belonging to people, other points falling on the background image belong to static obstacles. The background is obtained using Algorithm 1. First, each laser scan *S* is projected onto an occupancy-grid *X* using Algorithm 2. The values of cells that contain any laser point are set to 1, otherwise, 0 (line 6 in Algorithm 2). The n all projected grid maps X are summed up to obtain the total hit map *H* (line 7 in Algorithm 1), of which each element represents the hit count of the corresponding cell. To get the static background, we calculate the mean hit ratio by dividing *H* by *N*, where *N* is the total number of laser scans. If a cell’s hit ratio is greater than a predefined threshold, its value is set to 1, indicating it is occupied by a stable obstacle. Otherwise, its value is set to 0, indicating it is probably free of obstacles or occupied by moving obstacles. Once the background is given, the label of a point can be easily obtained according to their projected indices (line 17 in Algorithm 1), an example background of a dataset is illustrated in Figure 5. After obtaining the labels of all scans, neighborhoods of each point are extracted with a *k*D-tree. Since the class distributions are highly imbalanced in a dataset, we resample the negative examples to have the same size as positive examples.
**Algorithm 1** Automated Annotation of Laser Scans**Require:** set of laser scans S
**Ensure:** set of labels of laser scans Y
1:**function**AnnotateDataset(S)2: H←0, X←∅, Y←∅3: N←Length(S)4: **for** all S∈S
**do**5:  X←ProjectLaserScan(S)6:  Insert *X* into X7:  H←H+X        ▹ accumulate the hit count8: **end for**9: **for** all h∈H
**do**10:  **if**
h/N>threshold
**then**11:   h←1                ▹ static cell12:  **else**13:   h←0              ▹ nonstatic cell14:  **end if**15: **end for**16: **for** all S∈S
**do**17:  Calculate labels *Y* according to projected indices18:  Insert *Y* into Y19: **end for**20: **return**
Y21:**end function**
**Algorithm 2** Projection of laser points to a grid map**Require:** filtered laser points $P\in R_{+}^{2n}$
**Ensure:** grid map *M*
1:**function**ProjectLaserScan(*P*)2: M←03: r← the grid map’s resolution4: **for** all x,y∈P
**do**5:  i,j←⌊x/r⌋,⌊y/r⌋         ▹ projected index6:  $M_{j,i}\leftarrow 1$7: **end for**8: **return**
*M*9:**end function**


## 5. Obstacle Avoidance

### 5.1. Generation of Candidate Paths

The local path planner generates candidate paths by sampling laterally and longitudinally in the Frenet coordinate frame. Figure 6 shows a set of candidate paths with different lateral offsets and a single longitudinal offset. A candidate path is divided into three sections, the first section allows the robot to smoothly switch to the target lane, the second section of the path is parallel to the reference path and the last section allows the robot to reach the desired goal position.

The parameters of the local path planner are summarized in Table 1. $l_{i}$ refers to the latitudinal offset of each candidate path to the reference path, $s_{i}$ refers to the longitudinal offset at which a candidate path begins to be parallel to the reference path. The rate of change of a path is determined by $l_{i}/s_{i}$. Sampling in lateral and longitude dimensions allows the robot to choose paths with different steepness according to the size of obstacles and the distance from obstacles. For example, if the robot needs to bypass a nearby obstacle, it chooses a candidate path with a short longitudinal offset and a large lateral offset. The max deviation from the reference path is determined by $l_{max}$, which prevents the robot from entering into a configuration from which is difficult to come out. The $s_{min}$ and $s_{max}$ refers to the intervals of longitudinal sampling. The lateral sampling density Δl is set to 0.05 m, which is the same as the resolution of the cost map, and the longitudinal resolution Δs is set to 0.1 m. $d_{lookahead}$ refers to the maximum distance for evaluating the cost of a candidate path.

### 5.2. Path Selection

To find the best path to maneuver around obstacles, all the candidate paths that run over obstacles will be trimmed, the rest are evaluated by the following objective function *J*, which is the linearly weighted sum of three cost functions.
(1)$$\begin{array}{c} J=w_{1}J_{d}+w_{2}J_{t}+w_{3}J_{n}\\ J_{d}=max\left(\left\{cost_{i}\right\}\right)\\ J_{t}=\left|arctan2(l_{i},s_{i})\right|\\ J_{n}=\left\{\begin{array}{cc} 1 & \mathrm{if}\;l_{i}>0,\\ 0 & \mathrm{if}\;l_{i}\leq 0. \end{array}\right. \end{array}$$
where $\left\{cost_{i}\right\}$ is the costs of the grid cells that the robot covers along the path. $J_{d}$ is the cost of approaching obstacles, which makes the robot tend to maintain a larger distance from surrounding obstacles. The local cost map is utilized to efficiently calculate this cost, each cell of the local cost map has a cost value that decays with the distance to nearest lethal obstacles. $J_{d}$ is calculated as the maximum of the costs of cells that the footprint of the robot traverses along the path, as shown in Figure 7. $J_{t}$ is the deviation cost, which prevents the robot from deviating too much from the reference path to avoid obstacles. The last term $J_{n}$ is the social norm cost, which assigns higher costs to the paths on the left than those on the right side of the robot. This term encourages the robot to pass on the right, leading to socially compliant navigation. In summary, this objective function prefers paths that: (a) remain far from obstacles; (b) pass on the right side; (c) do not deviate too much to bypass obstacles.

### 5.3. Behavior States

A heuristic rule, which has been widely used in many automated guided vehicles (AGVs), is applied here in order to reduce unnecessary avoidances and improve the efficiency of navigation in dynamic environments, shown in Figure 8. Surprisingly, the strategy is not to avoid at all. That is, when the robot encounters an obstacle, it stops, warns people and waits for it to move away until a timeout event has been triggered. Experience tells that this simple approach has a good chance of success especially when the robot moves on the right side of a hallway. On e possible reason is that humans have more mobility, flexibility and smaller footprints than robots; they tend to give way to the robot when they understand its intention. However, this simple rule cannot handle all exceptional situations, such as encountering a person who is inconvenient to give way, an emergency patient or an object left unattended, entering a narrow door or an elevator. In these cases, the robot has to switch to obstacle avoidance mode and take slight deviations from the prescribed path to maneuver around the obstacles. Moreover, the local path planner is only executed when needed; this also saves computational resources.

### 5.4. Path Tracking

To track the reference path, a Pure Pursuit controller is employed to produce motion commands sent to the robot wheels. It runs at a frequency of 20 Hz and is used in the forward simulation to avoid collisions. The controller constantly simulates a virtual robot moving forward for a certain distance and checks whether the simulated trajectory is in collision with obstacles. If it is, the robot will stop and try to avoid the obstacle, otherwise, the robot will continue to travel along the path.

The pure pursuit controller is illustrated in Figure 9. The inputs of this algorithm are the reference path and the current robot position. The pure pursuit controller calculates the curvature of the circle, which connects the robot’s center and an intermediate goal point on the reference path ahead of the vehicle by a look ahead distance *L*. The curvature of the circle is given by κ=2sinα/L. For a given velocity *v*, the angular velocity is ω=2vsinα/L. If the angular velocity exceeds the physical limit, it will be truncated between $\left[-\omega_{max},\omega_{max}\right]$, then the velocity of the robot is reset to v=ω/κ. For smooth following behavior, the look-ahead distance is often scaled with the robot’s current translational velocity,
$$L=\frac{v}{v_{max}}\left(L_{max}-L_{min}\right)+L_{min}$$

The velocity commands of the left and right wheels of a differential robot are given by
$$\begin{array}{cc} v_{left} & =v-\frac{2}{B}\omega \\ v_{right} & =v+\frac{2}{B}\omega \end{array}$$
where *B* is the length of the wheelbase, $v_{left}$ and $v_{right}$ is the velocity of left and right wheel respectively.

If the angle between the target point and the robot heading direction is above a certain threshold, the deviation between the shortest straight-line path and the arc is large. In this case, the robot first rotates to align its heading with the virtual target point and then moves forward.

## 6. Experimental Evaluation

### 6.1. Experiments on People Detection

In this section, we compare the proposed method with the Dilated CNN, PointNet and the DROW detector on the dataset collected in crowded environments.

#### 6.1.1. Dataset

We recorded 18 sequences at several places (such as teaching buildings and canteens) of our school. The recording time of a dynamic dataset ranges from 15 to 30 min. An example scene is shown in Figure 10. The datasets are recorded with a smaller robot equipped with a Hokuyo laser range finder mounted at approximately 40 cm above the ground plane. All records are stored as rosbags and contain about 600 k raw laser scans. All dynamic datasets are annotated using Algorithm 1 with threshold 0.2. We take 4 dynamic datasets and 3 static datasets as the training set; others are used as the test set. Note that the test set covers different areas from the training set; this can measure the generalization performance of different approaches to unseen cases.

#### 6.1.2. Radius Size

The radius of the neighborhood determines the size of the receptive field of the network. Experiments were conducted to study the relationship between the radius size and performance. In Figure 11, the results show that if the radius is too small, the neighborhoods do not contain enough information for reliable detection, which decreases the accuracy. If the radius is too large, the neighborhoods contain much information of background objects, which makes the network prone to overfitting and decreases the ability of generalization to new cases. From the results of Figure 11, the radius between 1.2 m and 2 m is a suitable choice to distinguish between the background and people.

#### 6.1.3. Baselines

We compare the proposed method to three baselines: the dilated convolutional neural network (CNN), the distance robust wheelchair/walker (DROW) detector and PointNet. The evaluation metrics are the pixel accuracy ($\mathrm{pAcc}=\frac{TP+TN}{TP+FP+TN+FN}$), the Intersection over Union (IoUs) (${\mathrm{IoU}}_{people}=\frac{TP}{TP+FP+FN}$ or ${\mathrm{IoU}}_{no\_person}=\frac{TN}{TN+FN+FP}$), and the mean IoU $(\mathrm{mIoU}=\frac{1}{2}\left({\mathrm{IoU}}_{people}+{\mathrm{IoU}}_{no\_person})\right)$, where TP, FP, TN, FN are the number of true positive, false positive, true negative and false negative predictions respectively.

All experiments are conducted on a desktop computer with an Intel Core i9-9900 K @3.6 GHz CPU and a NVIDIA RTX2080Ti GPU. The Adam optimizer is used to minimize the cross-entropy loss function with a learning rate of 0.001 and a momentum of 0.98. All detectors are trained for 20 epochs with a batch size of 256.

We implemented a dilated CNN which consists of 8 convolutional layers with dilation factors 1,2,4,8,4,2,1,1. Its input is a binary occupancy grid centered on the robot with a size of 10 m × 10 m and a resolution of 0.1 m, its output is a score map of the same size and its loss function is masked using its input. We modified the open source DROW (github.com/VisualComputingInstitute/DROW) detector for the semantic segmentation task by removing the voting and nonmaximum suppression steps. We train the PointNet (github.com/fxia22/pointnet.pytorch) on our dataset with the default parameters.

#### 6.1.4. Results

The pixel accuracy and mean IoU scores of different approaches are presented in Table 2 the precision-recall curves are shown in Figure 12. The result shows that the CNN method overfits the training set and performs badly on the unseen dataset. Its computational cost is determined by the grid resolution of the input image. A large resolution results in loss of information and decreases the performance. The PointNet detector seems to have a high score on overall accuracy, but it is worst on detecting people. The reason is that most points in a laser scan are belong to static obstacles, so the global feature is dominated by the features of static points. The DROW detector has comparable performance to our method, both methods have similar precision-recall curve and achieves good performance on the test dataset. Since the dataset are balanced by resampling, both methods do not suffer from the class imbalance problem.

### 6.2. Experiments on Obstacle Avoidance

Comparison between different obstacle avoidance algorithms in real world is costly, and the source codes of most algorithms are not publicly available or tightly integrated into a big system. The most well known publicly available navigation package is ROS Navigation, which implements the A* and DWA algorithms. However, its behaviors are difficult to predict and not consistent in dynamic environments. In this section, we give a qualitative experimentation of the proposed algorithm.

To verify the performance of the proposed obstacle avoidance algorithm, several experiments are carried out with a real robot in the corridor in front of our laboratory (shown in the video). In all the experiments, the robot follows the same predefined path from the start position to the goal position. The maximum linear velocity of the robot is 1 m/s, the maximum lookahead distance of the local planner is 2.5 m. The individual terms of the objective function are normalized between [−1,1], the weights of each term are $w_{1}=1.0$, $w_{2}=0.9$, $w_{3}=0.5$, these weights are manually tuned in this experiment. Note that the paths that collide with obstacles are already removed, so these weights only affect the comfort of the selected path but should not make the robot collide with obstacles. The maximum waiting time before the local path planner is called to search for a feasible path is 2.0 s. Moreover, we conducted some experiments on the third floor of a building in Anhui Provincial Hospital.

#### 6.2.1. Case 1: Static Obstacles

In the first experiment (Figure 13A, three static obstacles are placed in the corridor. The trajectory and the velocity of the robot are shown in Figure 13B,C. In this case, the robot stops in front of the first obstacle, waits for a while, rotates to the right, switches to the lane on the right side of the robot, follows this lane to the third obstacle and then switches back to the reference lane. This experiment demonstrates that the local planner can drive the robot through static environments without colliding with obstacles.

#### 6.2.2. Case 2: Crossing Scenarios

In the second experiment (Figure 14), there are two people walking along the corridor. One person suddenly appears from the blind spot and crosses the path of the robot at a right angle. Another person comes from the opposite direction and obliquely crosses the path of the robot. In this situation, the robot stops and waits for the people to leave, and then continues to move straight. This is a reasonable behavior to stop and not to deviate from the straight path, since the total distance and time are the shortest.

#### 6.2.3. Case 3: Following a Person

In the third experiment (Figure 15), a person walks in front of the robot at a lower speed. When the robot catches up with the person, it stops and waits. When the person leaves far enough away, the robot continues to follow the same lane. In this experiment, the person deliberately does not give way to the robot, so the robot moves forward a little bit, stops and then moves forward, as shown in Figure 15C.

### 6.3. Experimental Results in the Hospital

We conducted several experiments in a building of the Anhui Provincial Hospital (2000 beds) to verify the performance of the proposed algorithm. This hospital has large traffic of people and a large volume of materials to be transported every day. According to statistics, more than 1.37 million intravenous infusions and 1.31 million specimens have been delivered manually in this hospital in 2018. The environment in this hospital is a challenge for robots because many of the places they pass through are crowded with people at some time, such as corridors and halls. Therefore, people detection and safely avoiding people become crucial for our application.

In the experiments, the task of the robot is to transport drugs from the central pharmacy to the nurse station on the third floor. Figure 16 shows the grid map and the topological map of the third floor of a building of the hospital. Since the robot cannot control elevators and doors, we will help the robot press elevator buttons and open doors during the experiments. During several days of experiments in the hospital, we found that people who do not carry heavy things often change paths to avoid the robot when they notice it, especially when it moves on the right side of the corridor. Figure 17A,B shows such a situation that a worker who pulls a trailer intuitively gives way to the robot ahead of time when he sees it. The algorithm does not produce wavy trajectories in all experiments; the behavior and the intention of the robot are easy to predict. Figure 17C shows that the people do not feel threatened when they pass by the robot, since they already understand the robot’s trajectory. There are also several situations that the robot cannot handle now. For example, if both people and the robot cannot go back and give way in narrow spaces, such as doors and elevators, then neither side can move forward.

## 7. Conclusions

This paper focuses on the development of two important components (i.e., people detection and obstacle avoidance) of a differential-drive robot for safe navigation in human-populated hospital environments.

For people detection, we proposed to firstly classify each proposal point using the information of its local neighborhoods with PointNet, due to its invariance to the permutations of the input set. Secondly, in order to reduce the number of points to be evaluated during inference, we use jump distance clustering method to split the laser scan into groups and select the centers of each group as proposal points. Since neural networks need a large dataset to train, we proposed an approach to automatically label dataset collected in actual environments. This method also reduces the cost of deploying laser-based people detection algorithms in new scenarios. Experimental results show that the proposed method achieves comparable performance on par with the state-of-the-art approaches.

For obstacle avoidance, we argued that the behaviors generated by the algorithm in human-populated environments should be legible and consistent. Pure reactive approaches can not be directly deployed in dynamic environments, since they have several drawbacks as mentioned above. To tackle these problems, we proposed to avoid obstacles by switching lanes generated by laterally sampling along the reference path. One advantage of the proposed method is that the candidate paths are consistent with the reference path and guaranteed to reach the goal position. Another one is that the robot’s behaviors can be intuitively understood by nearby pedestrians, this improves the safety of navigation. Moreover, we adopted a stop-wait-warn rule to reduce unnecessary deviations and improve the efficiency of navigating in dynamic environments. In addition, we proposed a new objective function which takes account of the distance to obstacles, the deviation from the global path and the social cost of passing on the right. Finally, experiments carried out in real-world environments confirm the effectiveness of the proposed method.

In future work, we consider adding a BACKWARD behavior to the behavior states of the robot. Since in some narrow places, the robot may need to give way to let the other side go through. In this situation, the robot needs to predict the path that the people will take using the information obtained from the surrounding environment. We will also make the behavior of the robot a little smarter, for example, adjusting the waiting time according to the types of the front obstacle and choosing lanes according to orientations and speeds of nearby people. 

## Figures and Tables

**Figure 1 sensors-21-00961-f001:**
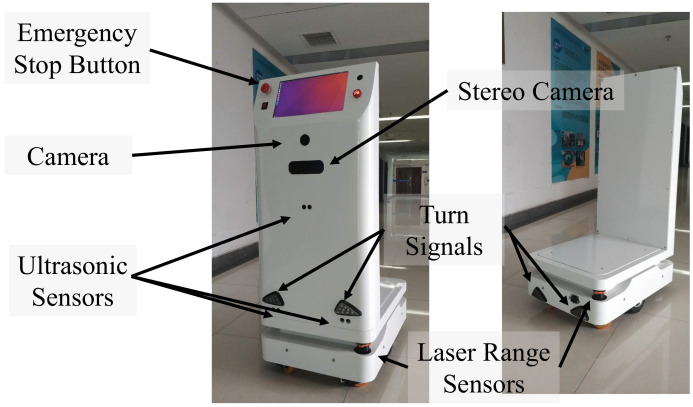
The prototype of our differential-drive transfer robot.

**Figure 2 sensors-21-00961-f002:**
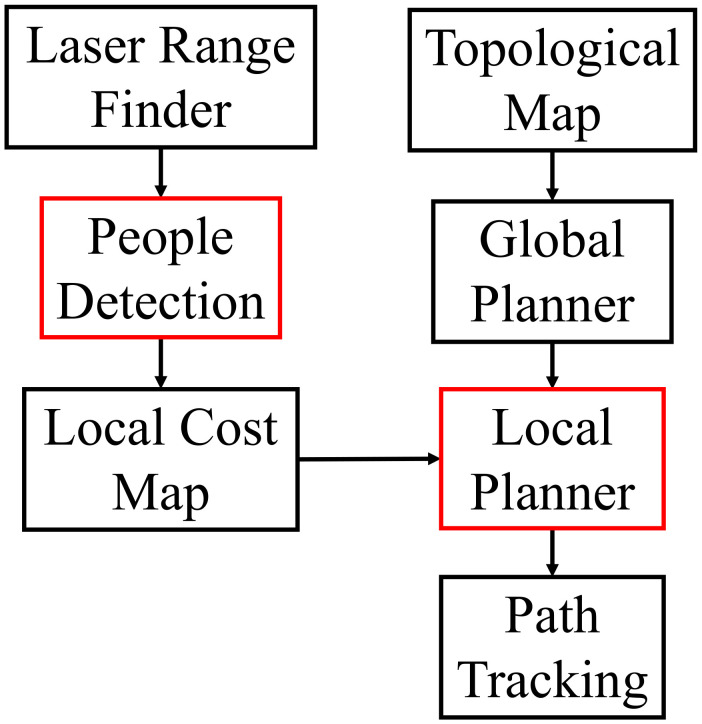
Overview of the main components of the software system. In this paper, we mainly focus on the laser-based people detector and the local path planner.

**Figure 3 sensors-21-00961-f003:**
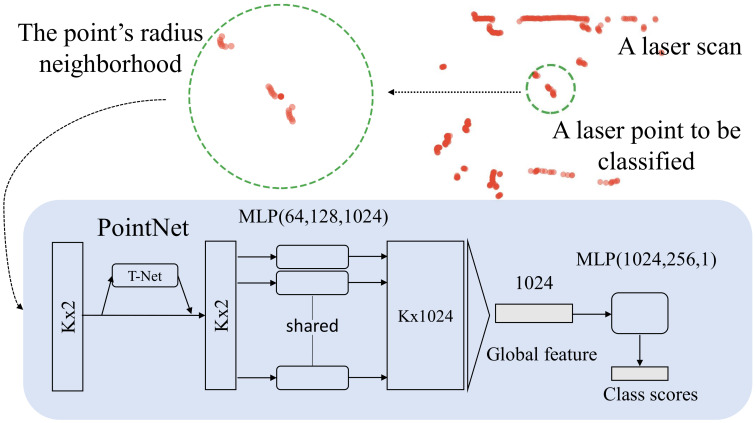
The neighborhoods of a set of proposal points are inputted to the PointNet to predict their scores. The neighborhoods are resampled to a fixed size *K* for parallel computing. “MLP” stands for multi-layer perceptron, numbers in brackets are layer sizes. The output is the probability of the class a point belongs to.

**Figure 4 sensors-21-00961-f004:**
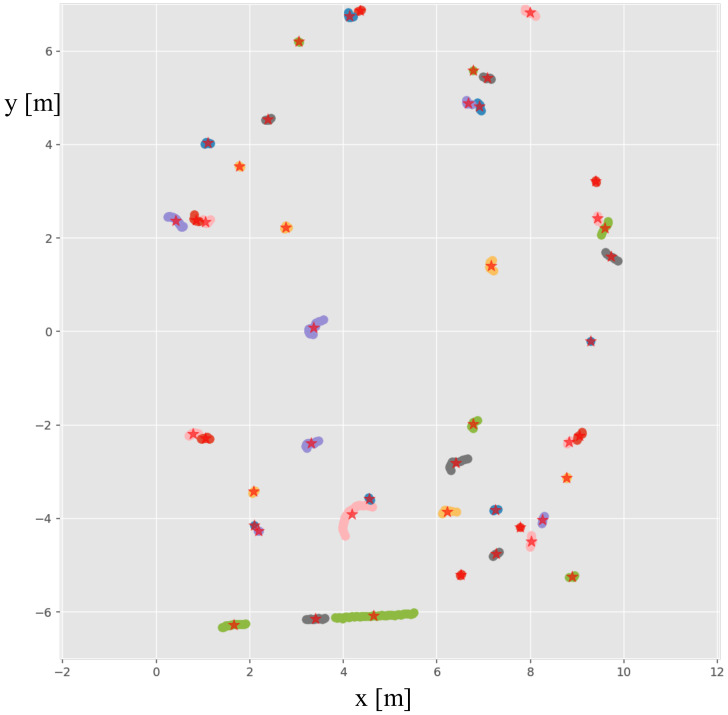
The jump distance algorithm groups a laser scan into clusters with different colors according to the difference of ranges of adjacent laser beams. The centers of each cluster are marked as ☆. The center of the laser is at the (0,0) coordinate. Only the laser points within 10 m away from the center of the robot are shown here.

**Figure 5 sensors-21-00961-f005:**
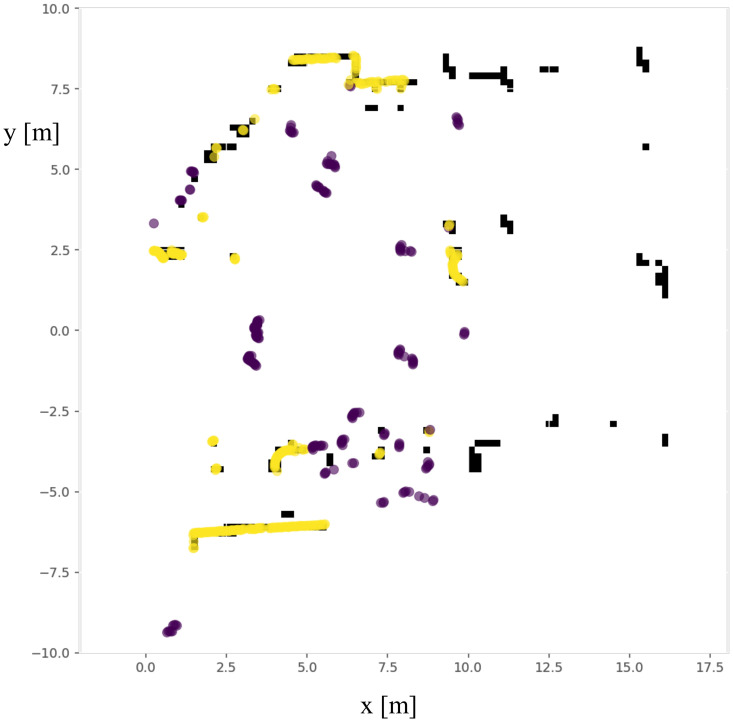
An example background (the black grid cell) generated with Algorithm 1 using a dynamic dataset collected in a real scenario. An example laser scan (colored dots) is projected on the background image. The yellow points which falls on black grids are static points, the purple ones are dynamic points.

**Figure 6 sensors-21-00961-f006:**
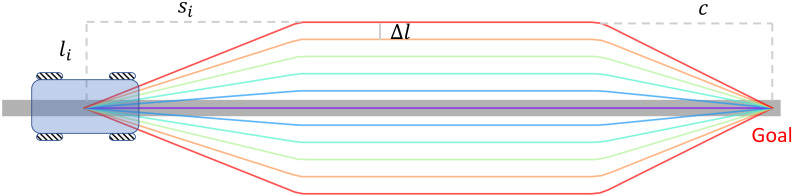
Example candidate paths generated by the local path planner with different lateral offsets and a longitudinal offset. The middle gray line is the reference path which are represented by a list of waypoints. The candidate paths are composed of three sections, the first and the last sections allow the robot to smooth switch paths. When current path is not valid, the local path planner considers several candidate paths (lanes) at once and chooses the best lane according to a loss function. Since all the lanes are determined by the reference path beforehand, the behavior of the robot can be easily predicted by surrounding people.

**Figure 7 sensors-21-00961-f007:**
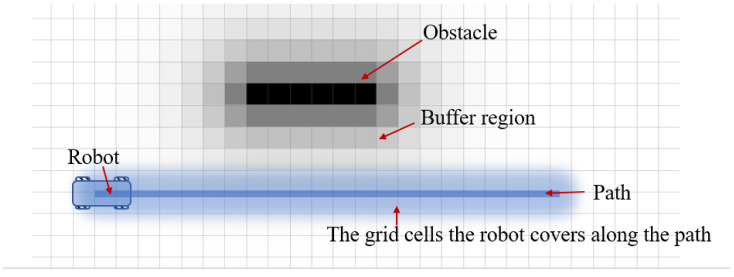
Calculation of the cost of a path to obstacles. The cost of the path is calculated as the maximum cost of the grid cells that the robot covers along the path.

**Figure 8 sensors-21-00961-f008:**
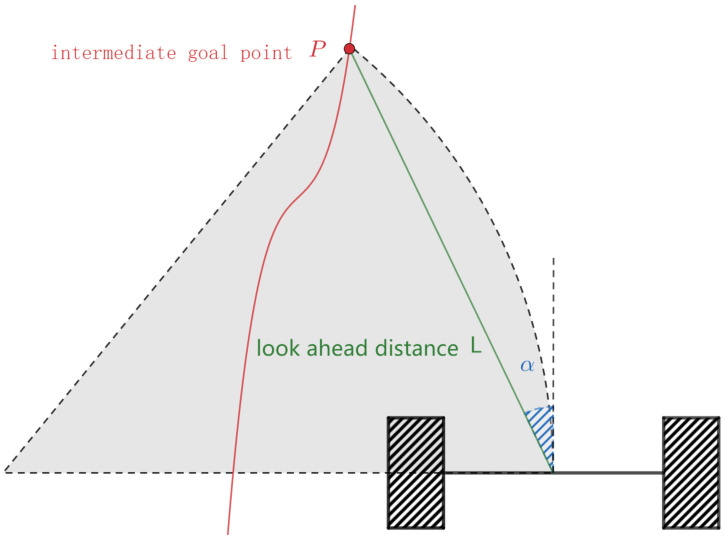
Illustration of pure pursuit technique. The red curve indicates the global reference path and the green line represents the one look ahead distance between the center of the differentially-drive robot and the intermediate goal point. The purpose of this algorithm is to calculate the curvature of the dotted circle.

**Figure 9 sensors-21-00961-f009:**
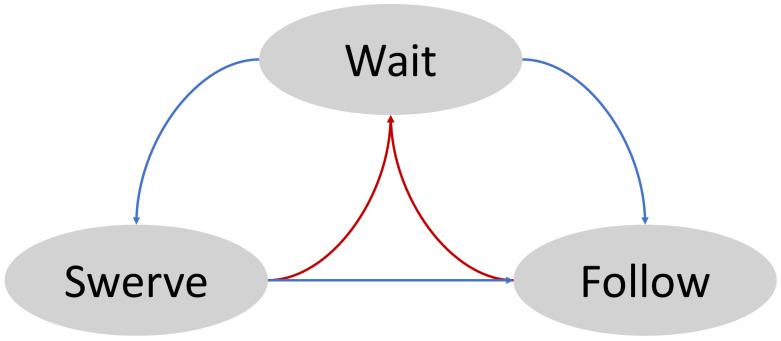
Behavior states of the local planner.

**Figure 10 sensors-21-00961-f010:**
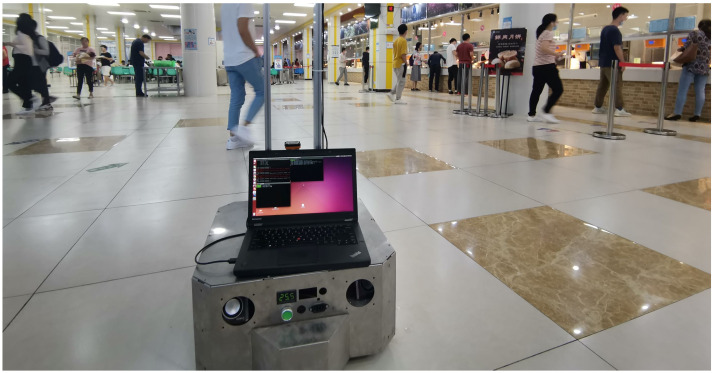
Collecting a dynamic dataset in a real-world environment.

**Figure 11 sensors-21-00961-f011:**
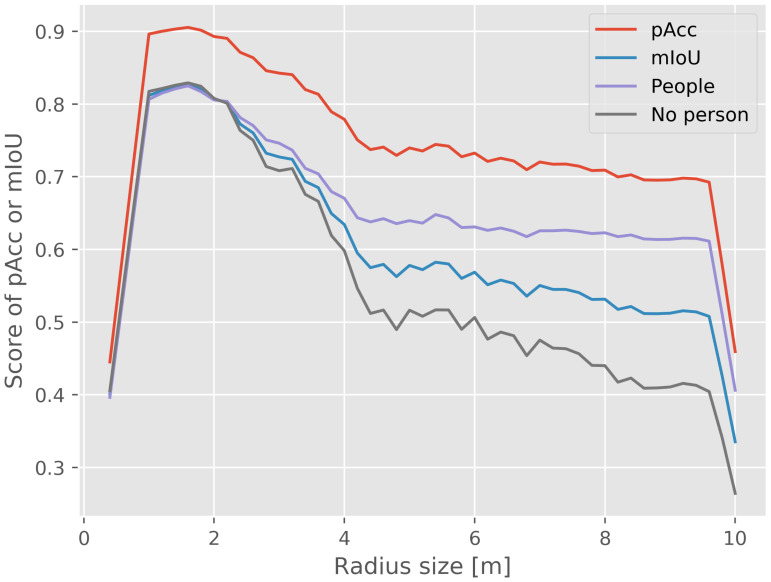
Effect of the neighborhood size *r* on the generalizability of the network. The colored curves show the scores of the network with different neighborhood radii on the test set. The radius between 1.2 and 2 m is beneficial for the generalizability of the model.

**Figure 12 sensors-21-00961-f012:**
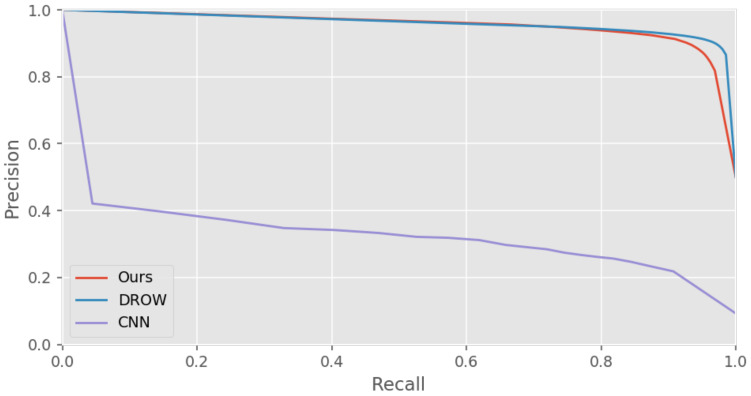
The precision-recall curve of DROW, a dilated CNN and our method on the collected dataset.

**Figure 13 sensors-21-00961-f013:**
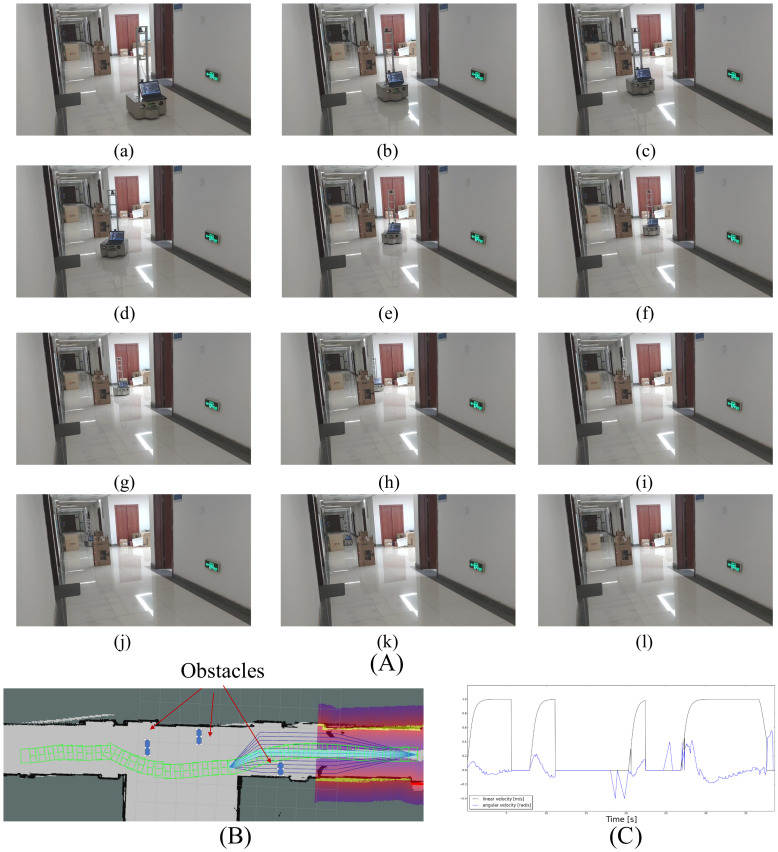
Case 1: the robot follows the reference path while avoiding three static obstacles. (**A**) The screenshots of the experiment environment and the trajectory of the robot. (**B**) The screenshot of the internal states. The local costmap is centered on the robot’s current position, the purple color means the cells have low cost. The green rectangles show the trajectory of the robot. (**C**) The linear and angular velocity of the robot, the *x* axis is time, the *y* axis is the linear (m/s) and angular velocity (rad/s) of the robot.

**Figure 14 sensors-21-00961-f014:**
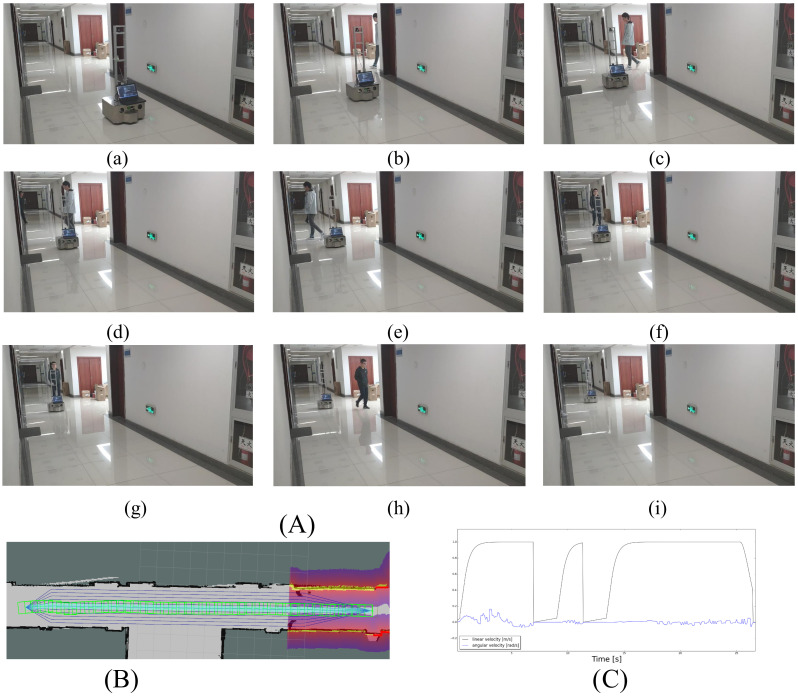
Case2: crossing scenarios. (**A**) Two persons crossed the reference the path of the robot, the robot stopped and waited a little while and continued to follow the reference path when they walked away. (**B**) The trajectory of the robot. (**C**) The linear and angular velocity of the robot.

**Figure 15 sensors-21-00961-f015:**
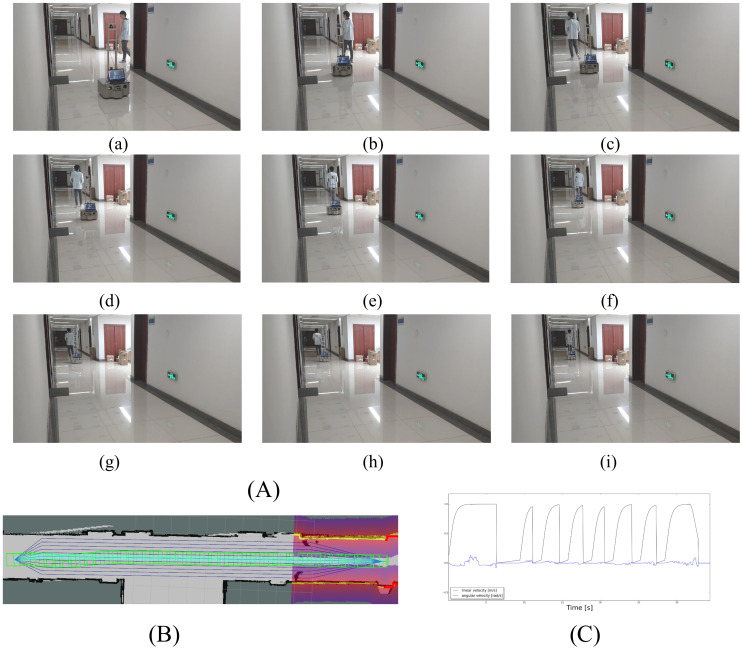
Case3: following a person. (**A**) A person walking in the front of the robot in the same lane. Since the speed of the robot is about the same as the person, it continues to follow the front person and does not change to another lane to surpass him. (**B**) The trajectory of the robot. (**C**) The linear and angular velocity of the robot.

**Figure 16 sensors-21-00961-f016:**
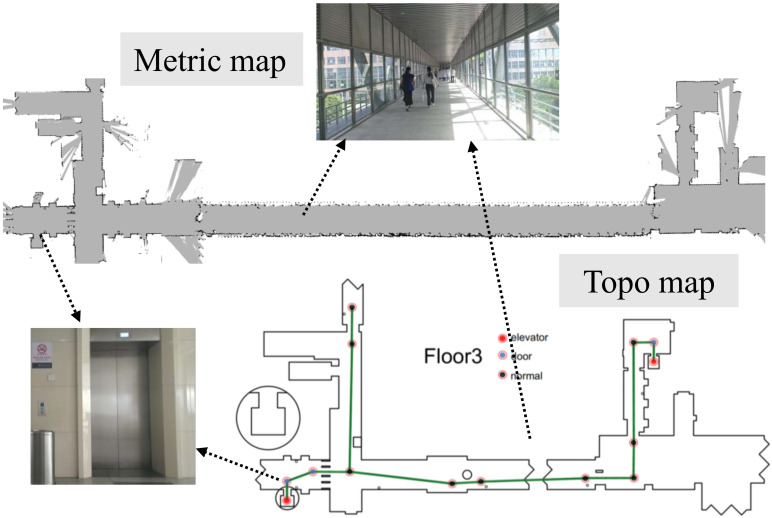
The occupancy grids map (**top**) and the topological map (**bottom**) of the third floor of the hospital. Grey indicates the areas free of obstacles. The nodes and paths of the topological map are manually drawn with a graphical tool. Elevator nodes indicate that they are in elevators. The door nodes indicate that the robot may need to stop and check whether the door is open before it can go through it. The normal nodes indicate that the robot does not need to stop on these intermediate nodes.

**Figure 17 sensors-21-00961-f017:**
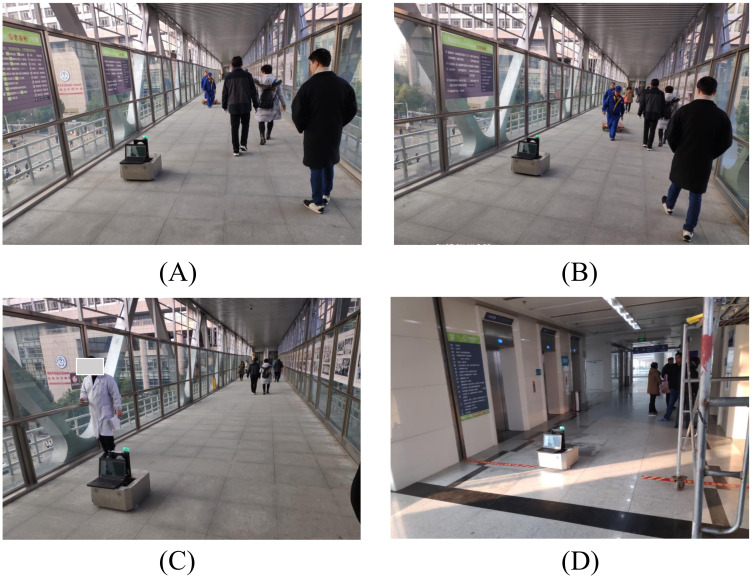
Experiments in Anhui Provincial hospital. The robot needs to safely navigate through corridors and halls populated with humans. The robot is passing through the long corridor (**A**–**C**) and the elevator hall (**D**) of the hospital.

**Table 1 sensors-21-00961-t001:** Parameters of the local path planning algorithm.

Parameter	Meaning
$l_{i}$	lateral offset of the *i*-th candidate path
$s_{i}$	longitudinal offset of the *i*-th candidate path
$l_{max}$	maximum lateral offset
$s_{min}$	minimum longitudinal offset
$s_{max}$	maximum longitudinal offset
Δl	lateral sampling density
Δs	longitudinal sampling density
$s_{c}$	longitudinal distance of convergence
$d_{lookahead}$	look-ahead distance

**Table 2 sensors-21-00961-t002:** Performance comparison of different approaches on the collected dataset.

Method	pAcc	mIoU	People	No Person
CNN	81.7	51.45	22.10	80.8
PointNet	90.3	45.2	0.0	90.4
DROW	91.7	85.1	85.3	84.9
Ours	91.8	85.8	85.8	85.7

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
