# Peer review of "Laser-Based People Detection and Obstacle Avoidance for a Hospital Transport Robot"

_sensors, 2021, doi:10.3390/s21030961_

Round 1
Reviewer 1 Report
Topic of the paper is very actual and presents interesting problem.
Introduction provides enough information in solved area of interest with relevant references.
Selected methods are adequate to solved problems. Paper has good theoretical backround.
Presented experimental results are very valuable and it will be also interesting for readers.
The conclusion summarizes the achieved results and clearly states the individual benefits of the solution. In the end, however, there is no plan for the future in research and development in this area.
I recommend to publish it
Author Response
Thank you for your feedback and positive comments
Point 1: The conclusion summarizes the achieved results and clearly states the individual benefits of the solution. In the end, however, there is no plan for the future in research and development in this area.
Response 1: In future work, we consider adding a BACKWARD behavior to the behavior states of the robot. Since in some narrow places, the robot may need to give way to let the other side go through. In this situation, the robot needs to predict the path that the people will take using the information obtained from the surrounding environment. We will also make the behavior of the robot a little smarter. For example, adjusting the waiting time according to front obstacle type, choosing lanes according to orientations and speeds of nearby people.
Reviewer 2 Report
The paper describes a robotic system for navigation in an indoor environment containing static and dynamic obstacles. The problem is well known in classic mobile robotics.
The paper is well presented and the introduction and motivation of the work are correct. However, in my opinion, there is not enough scientific contribution for publication as it is.
The work proposes a functional prototype that integrates a combination of available techniques. My main concern is that only a qualitative description of the system behaviour is provided. A much intense experimentation is required to assess system robustness, and give statistically significant measures of system performance in comparison with alternative methods.
Some other aspects that, under my opinion, should be clarified are the following ones:
- The robot includes two scan lasers covering 270 degrees, that means some overlapping is produced. Doesn't this generate an imbalanced hit estimation around the robot?
- The video shows a repetitive label change in some map areas. Shouldn't this be tackled in the system to assign a lower certainty to those regions?
- The robot behaviour seems to suffer from violent breaking. This could affect the integrity of transported materials, and also lead to non anticipated reaction from people following the robot.
Another minor details I have found in the document, including incorrect English expressions, are:
- "Traditional approaches comes" in line 47
- "unorder point sets" in line 141
- Variable name change in line 215
- "Goal to our approach" in line 217
- "placing the robot in a place" in line 260
- Figures missing units, for example 4 and 5
- Figure 6 variables, table 1 parameters and text description in 5.1 subsection need to be revised and unified.
- Jd formula is not included in line 301
- Precise train/test proportions in line 343
- Figure 10 y axis is misleading
- How have been w1, w2 and w3 values selected?
- Figure 12b doesn't show obstacle position
- What do normal dots represent in figure 16
- "as mentioned above" in line 442
Author Response
We truly appreciate your thoughtful comments and the detailed feedback, which helped us in revising the manuscript. We would like to address your concerns:
Point 1: The work proposes a functional prototype that integrates a combination of available techniques. My main concern is that only a qualitative description of the system behaviour is provided. A much intense experimentation is required to assess system robustness, and give statistically significant measures of system performance in comparison with alternative methods.
Response 1: We find two open-source 2D laser-based people detector available on Github, the ROS leg tracker (https://github.com/angusleigh/leg_tracker) and the DROW detector (https://github.com/VisualComputingInstitute/DROW) and Gandalf detector (https://github.com/TUI-NICR/gandalf_detector), However, it is too difficult to modify the leg_tracker for our application (laser scan segmentation), it needs multiple scans to determine the legs in the scan and is tuned to track legs. However, our task is to detect people from a single scan. For the (distance robust wheelchair/walker) DROW detector, we modify it for our task by removing the voting and non-maximum suppression steps. Since in the DROW paper, the author compares DROW with this detector, and the result shows that DROW surpasses this detector by a large margin, besides the source code is old and not easy to adjust for our application. So we only measure the performance with the DROW detector and PointNet and CNN.
For the obstacle avoidance algorithm, many local path planners are not publicly available or tightly integrated into a big system, it can be difficult to modify them for our application. We can find the open-source DWA local planner (http://wiki.ros.org/dwa_local_planner), In order to jump out local minima, this local path planner needs the A* algorithm to find a path. However, this pure reactive method is not designed to work well in human-populated environments, since they don’t assume a static environment and don’t consider the influence of the behavior of the robot on surrounding humans. However, give intense experimentation to assess system robustness is difficult since the input of this system is real-world environments, the configuration of dynamic objects has lots of freedom. It is dangerous and costly to conduct experiments in real-world hospital environments. However, in the future, we will conduct experiments in simulated environments, now the challenge is how to simulate the behavior of humans.
Point 2: The robot includes two scan lasers covering 270 degrees, which means some overlapping is produced. Doesn't this generate an imbalanced hit estimation around the robot?
Response 2: The data of the two scan lasers are merged together using the ROS {ira_laser_tools} (http://wiki.ros.org/ira_laser_tools) package, the basic workflow is that the positions of these two lasers are calibrated, the scans of these two lasers are transformed to point clouds, two point clouds are merged, then create a virtual laser to see the point cloud at the center of the robot. Put simply, it will be the same as using one laser with a field of view of 360 degrees. So this will not generate an imbalanced hit estimation.
Point 3: The video shows a repetitive label change in some map areas. Shouldn't this be tackled in the system to assign a lower certainty to those regions?
Response 3: The repetitive label change in some map areas is caused by surrounding people and occlusion, assign a lower certainty to those regions can be solution, but difficult to transfer to a new environment. The other solution is to smooth the labels using multiple laser scans, we will try to solve this problem in the future work. For some applications, such as localization in dynamic environments, if one want to remove dynamic points from the laser, and use only the static points to match the static map, in my opinion, this problem can be tolerated.
Point 4: The robot behaviour seems to suffer from violent breaking. This could affect the integrity of transported materials and also lead to non anticipated reaction from people following the robot.
Response 4: In this experiment, we don’t use a smoother stop function. We consider limiting the velocity of the robot according to its distance to surrounding people or static obstacles. In a real-world environment, I think the robot should use light signals to warn nearby people, so when they are aware of the presence of the robot, or even can predict the future movement of the robot, the safety of the system is increased.
Point 5 and Response 5: Another minor details I have found in the document, including incorrect English expressions, are:
- "Traditional approaches comes" in line 47
- "Traditional approaches come" in line 47
- "unorder point sets" in line 141
- "unordered point sets" in line 141
- Variable name change in line 215
- We modify the name error in line 215
- "Goal to our approach" in line 217
- "Goal of our approach" in line 217
- "placing the robot in a place" in line 260
- "Placing the robot in a site" in line 260
- Figures missing units, for example 4 and 5
- Add units to figure 4 and 5 and 10
- Figure 6 variables, table 1 parameters and text description in 5.1 subsection need to be revised and unified.
- The lateral parameter are unified
- Jd formula is not included in line 301
- Jd equals the maximum cost of the cells the footprint traversed along the path,since the mathematical formula is not intuitive,we don’t included in line 301
- Precise train/test proportions in line 343
- We take 4 dynamic datasets and 3 static datasets as the training set, other 11 rosbags are used as the test set.
- Figure 10 y axis is misleading
- Figure 10 is the score of the metrics of pixel accuracy and IoUs
- How have been w1, w2 and w3 values selected?
- These 3 parameters are hand-tuned in this experiment
- Figure 12b doesn't show obstacle position
- We mark the positions of the obstacles now
- What do normal dots represent in figure 16
- Normal dots means the robot does not need to stop on these intermediate nodes.
- "as mentioned above" in line 442
- The disadvantages of pure reactive local planners for our application is discussed in section 2.2
Reviewer 3 Report
The authors presented their approach to solving the navigation problem of the mobile robot in indoor environments populated with multiple people. The proposed solution focuses on detecting the people in the laser rangefinder data and planning such an ajectory, that is comfortable for people in the robots surrounding. For this purpose authors proposed the straightforward, yet effective method of going straight to the target, and, in case of the obstacle, just wait until it moves away. Authors also claim that they incorporated social rules in the navigation, but the only rule used is to move on the right side of the corridor. Authors should also take into consideration the human comfort, that they mention in the related works section (e.g. when moving around the person pick the path that will not cross the person's path).
Authors also should provide a better comparison with the existing methods. Even though in the related works section multiple different methods are mentioned, only two of them are selected for comparison. It would be beneficial to include more methods, even trained on the different data (some pre-trained networks should be available).
As the robot is equipped with the camera (even the stereo camera) authors should at least write why this data source is omitted in the algorithm. Although the camera looks only ahead of the robot, the visual information can be very useful when fused with the people detection results from the laser data.
Authors claim that the algorithm "can be straightforwardly extended to 3D laser data." (lines 14-15). This is not addressed in the article and should be either proved on some experimental data or removed from the claims.
Terms used in the experiments (pAcc, mIoU, Legs, NoPerson) should be explained shortly at the beginning of the section.
The article needs additional proof-reading, as there are missing articles here and there. There are also some missing/wrong words, e.g.:
Line 92, missing word? "Takahashi [8] ?here? the repulsive potential function..."
Line 115: blogs -> blobs
Line 141: unorder -> unordered
Line 208: nature -> natural
Author Response
We truly appreciate your thoughtful comments and the detailed feedback, which helped us in revising the manuscript. We would like to address your concerns:
Point 1: Authors also claim that they incorporated social rules in the navigation, but the only rule used is to move on the right side of the corridor. Authors should also take into consideration the human comfort, that they mention in the related works section (e.g. when moving around the person pick the path that will not cross the person's path).
Response 1: Now, we only consider one explicit social rule, that is move on the right side of the corridor, and two implicit rules, one is keeps a larger distance to nearby people than to static obstacles such as walls, in order to not make them feel threatened, another is don’t change directions frequently so people can predict the movement of the robot. Another is not mentioned in the paper but can be easily implemented is adjusting the speed according to the distance to surrounding people. We do need to take into considerations more social rules to make the system more safe and robot, such as pick the path that will not cross the person’s path, recognize the groups of people and don’t pass in their middle, understand the intention of people and adjust the behaviour of the robot. However, the challenge now is we are not able to robustly predict the trajectory of the people, if we can reliably detect the orientations of people, we can will incorporate the information into our local path planner.
Point 2: Authors also should provide a better comparison with the existing methods. Even though in the related works section multiple different methods are mentioned, only two of them are selected for comparison. It would be beneficial to include more methods, even trained on the different data (some pre-trained networks should be available).
Respone 2: We find two datasets that are publicly available. However, the dataset don not contain lots of humans in the scene, in human crowded environments, the occlusion problem is significant. Since now we can easily collect large amount of real-world data, we train these models on our dataset which are more challenge that these two datasets. We find two open-source 2D laser-based people detector available on Github, the ROS leg tracker (https://github.com/angusleigh/leg_tracker) and the DROW detector (https://github.com/VisualComputingInstitute/DROW) and Gandalf detector (https://github.com/TUI-NICR/gandalf_detector), However, it is too difficult to modify the leg_tracker for our application (laser scan segmentation), it needs multiple scans to determine the legs in the scan and is tuned to track legs. However, our task is to detect people from a single scan. For the (distance robust wheelchair/walker) DROW detector, we modify it for our task by removing the voting and non-maximum suppression steps. Since in the DROW paper, the author comparisons DROW with this detector, and the result shows that DROW surpasses this detector by a large margin, besides the source code is old and not easy to adjust for our application. So we only measures the performance with the DROW detector and PointNet and CNN.
Point 3: As the robot is equipped with the camera (even the stereo camera) authors should at least write why this data source is omitted in the algorithm. Although the camera looks only ahead of the robot, the visual information can be very useful when fused with the people detection results from the laser data.
Response 3: Thanks for your advice. The camera is useful for obtaining rich abundant information from the environment, we consider to incorporate that information in the future work. For now, since the visual information is easily affected by the light condition and range measurement is not as accurate as laser scanners. We have not use visual information in our current work.
Point 4: Authors claim that the algorithm "can be straightforwardly extended to 3D laser data." (lines 14-15). This is not addressed in the article and should be either proved on some experimental data or removed from the claims.
Response 4: Yes we removed it from the claims.
Point 5: Terms used in the experiments (pAcc, mIoU, Legs, NoPerson) should be explained shortly at the beginning of the section.
Response 5: Yes, we add the explanation of these metrics at the beginning of the section.
Point 6: Line 92, missing word? "Takahashi [8] ?here? the repulsive potential function..."
Line 115: blogs -> blobs
Line 141: unorder -> unordered
Line 208: nature -> natural
Response 6: Thanks, we have corrected the mistakes.
Reviewer 4 Report
in the paper, a wide range of well known signal processing tools are used to construct the algorithm of people detection and obstacle avoidance for a mobile robot that navigates in human-populated environment.
The review of the state of the art is fair and exhaustive.
The described algorithms are fair. The proposed solutions improve the safety of the robot's movement in the hospital environment. The Authors propose more sophisticated objective function and some new agorhytmic details in the strategy of a choice of paths modification. Automatic annotation of the samples can be also considered as an important contribution of the Authors. Efficiency of the solution is compared against some other algorithms. The experimental verifications are modest so far.
The paper can be published as an interesting description of the application of known tools of signal processing to the solution of a particular problem
A lot of editorial awkwardness should be corrected, for example:
In the segment starting with line 214 till 217 not all symbols are explained (l). Do the symbols in the Table 1. concern also the parameters in this segment?
One can guess the meaning of the notation „MLP(64, 128, 1024)” but it is not explained;
The term "A proposal point" is not clear;
All terms and symbols in Figure 3 should be explained;
in Figure 4 and in Figure 5 – axes are not described;
In Figure 10 the vertical axis is described using the abbreviations never decrypted in the paper;
Readability of the paper should be improved.
Author Response
We truly appreciate your thoughtful comments and the detailed feedback, which helped us in revising the manuscript. We would like to address your concerns:
Point 1: A lot of editorial awkwardness should be corrected, for example:
In the segment starting with line 214 till 217 not all symbols are explained (l). Do the symbols in the Table 1. concern also the parameters in this segment?
One can guess the meaning of the notation „MLP(64, 128, 1024)” but it is not explained;
The term "A proposal point" is not clear;
All terms and symbols in Figure 3 should be explained;
in Figure 4 and in Figure 5 – axes are not described;
In Figure 10 the vertical axis is described using the abbreviations never decrypted in the paper;
Readability of the paper should be improved.
Response 1: Thanks a lot. We add the explanation of in line 214. The symbols in the Table 1 does not concern the parameters in Section 4. MLP stands for multi-layer perceptron, i.e fully connected layers.
We modify ‘A proposal point’ to ‘A laser point need to be classified’ in Figure 3.
We add a description of the term in the caption of Figure 3.
We add the axes to Figure 4 and Figure 5
We change the label of the y-axis to “score of pAcc or mIoU” to remove the ambiguity.
Round 2
Reviewer 2 Report
The authors have answered the posed questions and corrected some mistakes.
In my opinion, a few aspects could be further developed before publishing.
Regarding the lack of statistical support, I understand the difficulties of obtaining extensive experimental results. However, I assume some repetitions of every case have been recorded. It would be interesting to include some in the paper, to get at least some qualitative impression of system robustness.
Since the weights w1, w2 and w3 have been manually adjusted, this should be commented in the paper. Also, a description of how sensible is the system to the variation of those values could be of interest to readers.
The formula for Jd function is not included citing it's "not intuitive". Is it there at least a paper describing that function that could be referenced?
Some figures (10, 12) have misplaced/unaligned text.
Author Response
We truly appreciate your thoughtful comments and the detailed feedback, which helped us in revising the manuscript. We would like to address your concerns:
Point 1: Regarding the lack of statistical support, I understand the difficulties of obtaining extensive experimental results. However, I assume some repetitions of every case have been recorded. It would be interesting to include some in the paper, to get at least some qualitative impression of system robustness.
Response 1: Evaluating different obstacle avoidance methods is one of the most difficult. Evaluation of different algorithms is greatly affected by the testing environments. A fair and objective comparison is challenging. The metrics like distance, time is not enough to evaluate the robustness of different algorithms. The purpose of this work is to develop a simple and practical algorithm to avoid dynamic obstacles (people) in real-world environments. While general obstacle avoidance algorithms focus on computing a feasible path or velocity commands to avoid a static obstacle but don’t consider the interactivity of people and the legibility of the path. The simplicity of the path that the robot takes is important for nearby people to understand and react to the robot. Since this algorithm is heuristic and only a few factors are considered, for the time being, the smartness of this algorithm is limited.
Point 2: Since the weights w1, w2 and w3 have been manually adjusted, this should be commented in the paper. Also, a description of how sensible is the system to the variation of those values could be of interest to readers.
Response 2: Yes, we add a description of the weights. Since the paths that collide with obstacles are already removed, so these weights only affect the comfort of the path the robot selected, but should not make the robot collide with obstacles. So one can safely set the w1, w2, and w3 to 1 to consider all these three cost terms. However, the smartness of the behavior is difficult to evaluate.
Point 3: The formula for Jd function is not included citing it's "not intuitive". Is it there at least a paper describing that function that could be referenced?
Response 3: We add a formula to Jd function as J_d = max(\{cost_{i}\}) and a figure to explain how this cost is calculated.
Point 4: Some figures (10, 12) have misplaced/unaligned text.
Response 4: We modify the misplaced text in figures (10, 12)
limitedsmall in range or scopeMore (Definitions, Synonyms, Translation)
Reviewer 3 Report
Most of my concerns were addressed by the authors. The authors wrote in their answer why there is no better comparison of the existing approaches, but I think it should be also (at least partially) covered in the article.
Author Response
We truly appreciate your thoughtful comments and the detailed feedback, which helped us in revising the manuscript. We would like to address your concerns:
Point 4: there is no better comparison of the existing approaches, but I think it should be also (at least partially) covered in the article.
Response 4: We add an explanation of why some algorithms are not included in the first paragraph of section 6.2. “Comparison between different obstacle avoidance algorithms in real-world is costly, and the source codes of most algorithms are not publicly available or tightly integrated into a big system. The most well known publically available navigation package is \href{https://github.com/ros-planning/navigation}{ROS Navigation}, which implements the A* and DWA algorithms. However, its behaviors are difficult to predict and not consistent in dynamic environments. In this section, we give qualitative experimentation of the proposed algorithm.”
qualitativeinvolving distinctions based on qualitiesMore (Definitions, Synonyms, Translation)